# TOWARDS A RELIABLE AND ROBUST DIALOGUE SYSTEM FOR MEDICAL AUTOMATIC DIAGNOSIS

## ABSTRACT

Dialogue system for medical automatic diagnosis (DSMAD) aims to learn an agent that mimics the behavior of a human doctor, i.e. inquiring symptoms and informing diseases. Since DSMAD has been formulated as a Markov decision-making process, many studies apply reinforcement learning methods to solve it. Unfortunately, existing works solely rely on simple diagnostic accuracy to justify the effectiveness of their DSMAD agents while ignoring the medical rationality of the inquiring process. From the perspective of medical application, it's critical to develop an agent that is able to produce reliable and convincing diagnosing processes and also is robust in making diagnosis facing noisy interaction with patients. To this end, we propose a novel DSMAD agent, INS-DS (Introspective Diagnosis System) comprising of two separate yet cooperative modules, i.e., an inquiry module for proposing symptom-inquiries and an introspective module for deciding when to inform a disease. INS-DS is inspired by the introspective decision-making process of human, where the inquiry module first proposes the most valuable symptom inquiry, then the introspective module intervenes the potential responses of this inquiry and decides to inquire only if the diagnoses of these interventions vary. We also propose two evaluation metrics to validate the reliability and robustness of DSMAD methods. Extensive experimental results demonstrate that INS-DS achieves the new state-of-the-art under various experimental settings and possesses the advantages of reliability and robustness compared to other methods.

## 1 INTRODUCTION

Dialogue system for medical automatic diagnosis (DSMAD) aims to learn an agent to collect patient's information and make preliminary diagnosis in an interactive manner like a human doctor. This task increasingly grasps the attention of researchers because of its huge industrial potential (Tang et al., 2016). Similar to other task-oriented dialogue tasks (Lipton et al., 2018; Wen et al.; Yan et al., 2017; Lowe et al., 2015), DSMAD is composed of a sequence of dialogue-based interactions between the patient and the agent, which can be formulated as a Markov decision process and resolved by reinforcement learning (RL) (Mnih et al., 2015; Van Hasselt et al., 2016). Although several frameworks have been proposed (Xu et al., 2019; Wei et al., 2018; Peng et al., 2018; Tang et al., 2016), DSMAD is still far from being applicable, because these works only evaluate the agent based on the accuracy of diagnosis, but ignoring the importance of robustness and reliability for practical medical applications. The two major shortcomings of the current DSMAD methods are summarized below.

**Unreliable symptom-inquiry and disease-diagnosis**. It is reasonable to measure DSMAD by diagnosis accuracy since the accuracy is the ultimate goal of the task. However, in unilateral pursuit of high accuracy, DSMAD agent pays less attention to the rationale of the diagnosis process, reducing the trust of users. For example, a DSMAD agent might jump into a conclusion without inquiring about any symptom. As long as the diagnosis is correct, such an agent will still get a positive reward. In this sense, the correctness of diagnoses is not sufficient to reflect the performance of DSMAD, and might lead the agent to make a hasty diagnosis without interactions. Moreover, DSMAD should learn to make consistent disease-diagnoses according to the symptom-disease relation in the training data, insensitive to the noise happened during training.

**Sensitive to small disturbance**. Almost all of the current DSMAD methods combine the operation of symptom-inquiry and disease-diagnosis together and allow models to make the sequential deci-

sion in a black-box manner (Zhang & Zhu, 2018; Koh & Liang, 2017) without regulations, resulting in a system vulnerable to the noise during the interaction process. If we place one of the inquired symptoms to the self-report (to ensure the information is consistent between two cases), the agent sensitive to noise would make different diagnoses.

To this end, we propose a novel DSMAD agent, Introspective Diagnosis System (INS-DS) (Fig. 1) and two new evaluation metrics in terms of reliability and robustness. The diagnosis logic of INS-DS draws on the introspective decision-making process of human doctors. In real life, human doctors come into a conclusion if they believe that more inquiries make no difference. In INS-DS, the inquiry module is responsible for selecting the most valuable symptom to be inquired about, while the introspective module intervenes the potential answers of this inquiry to decide whether to inquire the symptom or inform the disease. Specifically, the introspection module assigns different possible answers to the inquiry resulting in multiple one-step-look-ahead dialogue states. Then it inspects whether the diagnosis results of these states are going to be varied. If the predicted results are the same, which means that inquiring the most valuable symptom inquiry would result in the same diagnosis, INS-DS will inform the disease instead. Otherwise, the agent would inquire about the symptom. Such mandatory introspection makes the inquiry more disease-related because the agent is not allowed to make a diagnosis until the agent has collected sufficient symptom information. It also makes the disease-diagnosis more consistent because the disease can only be informed when the comprehensive hypothesis test is passed.

To quantify the reliability and robustness of DSMAD, we also propose two novel evaluation metrics, namely, reliability and robustness.

**Reliability.** The purpose of the reliability metric is to quantify how confident the diagnosis is from the perspective of the model (internal trust or Int.) and user (external trust or Ext.). The internal trust is to testify whether the diagnoses made by the model is insensitive to the task-irrelevant factors, e.g., the sampling noise and parameter initialization. Specifically, for Int., we adopt the expected diagnostic probability of a set of bootstrapping models. These bootstrapping models are initialized with different parameters and trained on re-sampled data with replacement, to reduce the effect of the parameter initialization and the data sampling. Therefore, the higher Int. is, the less sensitive the diagnosis result is to the noise in the training process. The external trust is proposed to indicate the trust degree of a diagnosing process to users. Intuitively, patients are more likely to believe in the diagnosis made from the agents who request symptoms like human doctors. According to this intuition, for Ext., we compute the symptoms overlap ratio based on the co-occurrence between symptoms and diseases in the diagnostic dialogue dataset. The higher the level of Ext., the more likely the agent is to inquire symptoms like a human doctor.

**Robustness.** As for robustness, we draw on the inspiration from the noted adversarial attack (Kurakin et al., 2018) in machine learning models, which generally uses samples with a subtle modification that is indistinguishable for human but may be different for a machine to prove the vulnerability of a model. Our evaluation metric for the robustness is the final unaltered proportion of correct diagnoses after feeding the model with the attack samples constructed according to the formulas in Sec. 5.

The extensive experimental results evidence that INS-DS achieves the superior performance compared to other DSMAD baselines under various settings and possesses the advantages of reliability and robustness. We also conduct human evaluations on our INS-DS and achieve significant improvement in the aspects of diagnosis validity, symptom rationality as well as topic transition smoothness.

## 2 RELATED WORK

The task-oriented dialogue system is designed to accomplish specific tasks, like the ticket, restaurant booking, online shopping etc. (Lipton et al., 2018; Wen et al.; Yan et al., 2017). Most of the current task-oriented dialogue systems adopt the framework of reinforcement learning (Mnih et al., 2015; Lipton et al., 2018; Li et al., 2017), and some adopt the sequence-to-sequence style for dialogue generation (Madotto et al., 2018; Wu et al., 2019; Lei et al., 2018).

For medical dialogue system, due to a large number of symptoms, reinforcement learning is a better choice for topic selection (Tang et al., 2016; Kao et al., 2018; Peng et al., 2018). Tang et al. (2016) apply Deep Q-Network (DQN) (Mnih et al., 2015) to diagnose using synthetic data. While Wei et al. (2018) first did experiments on real-world data using DQN. To include explicit medical inductive bias for improving the diagnostic performance, Xu et al. (2019) proposed an end-to-end model

guided by a symptom-disease knowledge graph. KR-DQN applies the predefined conditional probability of symptom and disease to transform the Q-values estimated by DQN. However, it's often difficult to get a knowledge graph in real life. Moreover, most of these methods integrate symptom-inquiry and disease-diagnosis actions into one single reinforced policy network without considering the essential difference between symptom-inquiry and disease-diagnosis, allowing the agent to jump into conclusions rashly to avoid the possible penalty in the diagnosing process. Different from these methods, our approach divides the actions for symptom-inquiry and disease-diagnosis into two separate but cooperative neural modules. The idea of drawing the lesson from human is not new. Ling et al. (2017) improve the performance of diagnosis for clinical documents by replicating the memory recall and attention mechanism in the human decision-making process. They selected evidence from the external clinical source according to the hint sentences of the given clinical document and then applying reinforcement learning to filter out the useless evidence. Different from DSMAD, instead of interacting with the patient, their RL agent is used to decide whether the matching evidence is useful or not. And similar to (Xu et al., 2019) and (Wei et al., 2018), Ling et al. (2017) also rely on the agent to choose to stop but not through an introspective stop mechanism as ours.

As for evaluation, aware of that only final diagnosis accuracy is not sufficient, current methods measure the performance of the diagnosing process by computing the hitting rate of the inquired symptoms. Lei et al. (2018); Madotto et al. (2018); Wu et al. (2019) compute the entity F1 scores. Xu et al. (2019) use the matching rate compared to all asked symptoms while Tang et al. (2016); Kao et al. (2018); Peng et al. (2018) measure by the average number of queried positive symptoms of each user goal. However, the recorded symptoms in one user goal are generally a small subset of the actual disease-relevant symptoms in the whole dataset. Therefore, measuring DSMAD according to the recorded symptoms in each user goal could not reflect the actual performance of DSMAD in general. Different from them, one of our proposed metrics in this paper, namely, external trust, measures the matching rate according to the symptom-disease co-occurrence of the whole dataset. Besides, we also propose the metrics for evaluating the model's confidence in the decision and the robustness to the noisy input.

## 3 PRELIMINARY

### 3.1 DIALOGUE SYSTEM FOR MEDICAL AUTOMATIC DIAGNOSIS

Medical automatic diagnosis is to interact with the users, actively collect useful information, and conduct preliminary diagnosis according to the collected information. In this task, a user goal in the dataset includes a self-report (e.g., "Doctor, my child has a headache. What happened to him?"), a set of explicit symptoms, a set of implicit symptoms and a ground-truth disease. Usually, a patient simulator is built upon user goal to create an interactive environment. Given a user goal, the patient simulator starts the dialogue with the self-report, and then answers the inquiries from the DSMAD agent according to the recorded symptom information in the user goal. DSMAD agent chooses to inquire the next symptom or inform the disease according to the dialogue state in each turn. When the agent inquires a symptom, it would get meaningful feedback if the symptom is presented in the explicit symptoms or implicit symptoms of the user goal. And if the agent chooses to inform a disease, the discourse will be ended and the agent will receive feedback on the ground-truth disease. In this paper, we adopt the same patient simulator used in (Xu et al., 2019).

Generally, DSMAD consists of three elementary components: 1) natural language understanding (NLU), which is used to parse the language input of the patient to fill the semantic symptom slots; 2) dialogue management (DM), which is used to track the dialogue history and predict next dialogue state according to the gathered information; 3) natural language generation (NLG), which is used to generate natural language for either symptom inquiry or disease diagnosis. Different from works (Xu et al., 2019; Wei et al., 2018), this paper focuses on the decision-making part in DM, that's, using the reinforced agent to either inquire a symptom or inform a disease. The NLU and NLG components used in this paper are the same as (Xu et al., 2019).

### 3.2 NOTATION

We design our agent in deep reinforcement learning framework (Mnih et al., 2015). The medical automatic diagnosis can be formulated as a finite-horizon Markov decision process (MDP), defined by the tuple $(\mathcal{S}, \mathcal{A}, p, r)$, where the state space $\mathcal{S}$ and the action space $\mathcal{A}$ are discrete. Patient communicates with the dialogue system by *request+disease* (inform the self-report) or *confirm/deny/not-sure+symptom*. There are three types of symptom states, *positive*, *negative* and *not-sure*, represented by value 1, -1 and 0 respectively. We also provide a binary indicator for each symptom to indicate

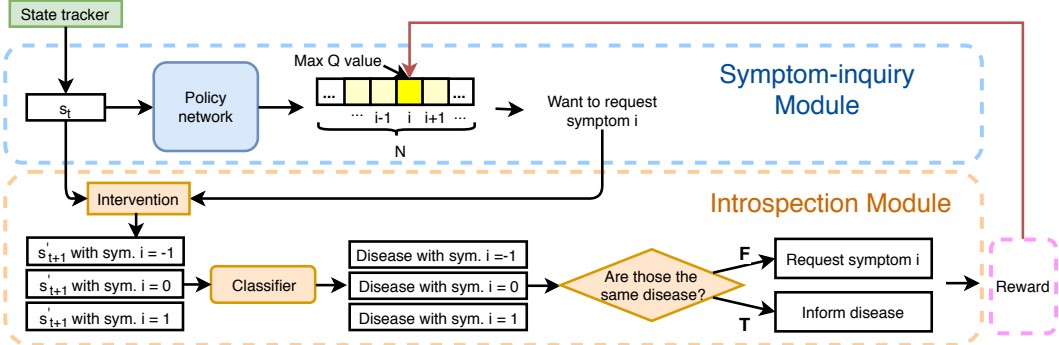

Figure 1: INS-DS with symptom-inquiry and introspection module. In the symptom-inquiry module, the current dialogue state $s_t$ is fed to the policy network to choose a candidate symptom $i$. Then, in the introspective module, $s_t$ is further intervened by setting the value of symptom $i$ with all possible values to generate multiple one-step-look-ahead $s'_{t+1}$ s. These $s'_{t+1}$ s are forwarded to the disease classifier to predict the diseases. The introspection module will choose to inform the disease if all predicted diseases are the same, otherwise, the agent will continue to inquire about the symptom $i$.

whether a symptom has been mentioned. At each turn, the dialogue state $s \in \mathcal{S}$ is composed of the value and the binary indicator of each symptom. The state tracker in DM records mentioned symptoms in the dialogue and produces the up-to-date dialogue state $s_t$ at time $t$. And the action space consists of the actions of symptom-inquiries and disease-diagnoses. Without causing conflict, $M$ denotes the number of the diseases and $N$ denotes the number of the symptoms. Therefore the size of the action space is $N + M$. The unknown state transition probability $p : \mathcal{S} \times \mathcal{S} \times \mathcal{A} \to [0, \infty)$ represents the probability of the next state $s_{t+1} \in \mathcal{S}$ given the current state $s_t \in \mathcal{S}$ and action $a_t \in \mathcal{A}$. The patient simulator emits a bounded reward $r : \mathcal{S} \times \mathcal{A} \to [r_{\min}, r_{\max}]$ on each transition.

The target of reinforcement learning is to maximize the expected accumulated rewards $\mathbb{E}[\sum_{t=0:T} \gamma^t r_t]$, where $\gamma$ is the discount factor to adjust the horizon of the foresight. In this paper, we adopt the classic q-learning to optimize the policy with parameter $\theta$, via minimizing the temporal-difference:

$$\min_{\theta} \mathbb{E}_{(s_t, a_t, r_t, s_{t+1})} \left[ \left( Q(s_t, a_t; \theta) - r_t - \gamma \max_a Q(s_{t+1}, a; \theta) \right)^2 \right]. \tag{1}$$

## 4 INTROSPECTIVE DIAGNOSIS SYSTEM

Generally, disease-diagnosis is the most critical step because only the correct diagnosis result could make the patient get the correct treatment. With a sense of security and responsibility, doctors introspect their decisions by imagining all the potential outcomes (Rubin, 1974) before making the final diagnoses. Inspired by this diagnosing process, we propose an introspective diagnosis system (INS-DS), which includes a symptom-inquiry module to select a symptom to be inquired about, and an introspection module to determine whether to inquire about that symptom or to inform the predicted disease. Different from the most popular diagnosis systems which treat symptom-inquiry and disease-diagnosis equally (Xu et al., 2019; Wei et al., 2018), our method only allows the disease-diagnosis to be made after considering all possible future outcomes in order to make the decision more reliable and interpretable. Also, INS-DS requires more correlation between the symptom-inquiry and the disease-diagnosis. In INS-DS, only when the candidate symptom can trigger different diagnosis results will the symptom be inquired about. And only when the most valuable inquiry proposed by the symptom-inquiry module makes no difference to the future diagnosis will the predicted disease be informed. INS-DS pipeline is illustrated in Fig. 1. Next, we elaborate the details of the two modules.

**Symptom-inquiry module.** At time $t$, the state tracker gathers historical symptom information to generate the current dialogue state $s_t$. Then $s_t$ is input to the symptom-inquiry module, which is responsible for selecting the most valuable symptom inquiry. As mentioned previously, $s_t$ consists of two vectors, one for the presences of symptoms and another for the visitation indicators. The two vectors are fed into two different Multi-Layered Perceptrons (MLPs) with parameters $\theta$, whose outputs are fused by Hadamard multiplication to produce the final Q values of all possible inquiries, i.e. $q_t \in R^N$. Both MLPs consist of two fully-connected layers. And only the symptom inquiry

with the maximum Q value will be selected as a candidate action and then be forwarded to the introspection module.

**Introspection module.** The introspection module would examine the necessity of the symptom inquiry by predicting potential diseases caused by all possible answers of the symptom inquiry. If the potential diseases are different, meaning that the symptom will cause differences in determining the disease, the agent would choose to request the symptom. Otherwise, the agent will inform the disease. Specifically, after receiving the most valuable symptom $i$ from the symptom-inquiry module, the introspection module intervenes all possible outcomes if the inquiry were made. This process is named as *Intervention* demonstrated in Fig. 1, which is an important ability for our Homo sapiens ancestors to achieve global dominion (Pearl, 2018). To do so, the intervention function replaces the symptom value of symptom $i$ in $s_t$ with all possible values (i.e., -1, 0 and 1) and sets the visitation indicator of the symptom as 1, to produce several imaginary states $s'_{t+1}$s (with the same size as the number of the possible answers to the inquiry). These one-step-look-ahead states are then forwarded to a disease classifier to produce their corresponding diagnoses. The classifier is an MLP with three fully-connected layers.

**Training objective.** In the introspection module, we adopt a classifier to infer the diseases of potential outcomes of intervention. The classifier is trained with input-label pairs $(s_t, d_T)$s for $t \in [0, T]$ collected after each training episode with length $T$ using cross-entropy. In order to optimize the inquiry policy, we employ Deep Q-learning following (Mnih et al., 2015), and two important DQN tricks, i.e. target network $Q'$ with parameter $\theta'$ and experience relay are adopted (Van Hasselt et al., 2016). In order to combine the introspection with the policy optimizing, the objective of INS-DS is derived from Equ. 1.

We use $Q(s_t, a_t; \theta)$ to denote the expected discounted sum of rewards, after taking an action $a_t$ under state $s_t$. Different from traditional temporal difference whose proposed action is the actual executed action, our policy with introspection is updated according to

$$\min_{\theta} \mathbb{E}_{(s_t, a_t, s_{t+1})}[(Q(s_t, a_t; \theta) - r(s_t, \text{Introspection}(s_t, a_t)) - \gamma \max_a Q'(s_{t+1}, a; \theta'))^2], \quad (2)$$

where $\theta'$ is the parameters of the target network updated by $\theta' \leftarrow \alpha\theta + (1 - \alpha)\theta'$, where $\alpha$ is the polyak factor. $\text{Introspection}(s, a)$ denotes the process in the introspection module. We use $\epsilon$-greedy exploration at training phase for exploration, selecting a random action according to probability $\epsilon$. We store the agents experienced transition at each time-step to a experience replay buffer.

## 5 EVALUATION CRITERIA

Most diagnosis systems use accuracy to evaluate performance (Wei et al., 2018; Xu et al., 2019). However, we argue that only accuracy is insufficient to evaluate a diagnosis system as a medical application. In particular, two major shortcomings observed in prior works (Wei et al., 2018; Xu et al., 2019) are the lack of reliability and robustness. To this end, we propose two novel metrics, namely, *reliability* and *robustness*, as complements to accuracy.

### 5.1 RELIABILITY

In real life, prudent doctors tend to be cautious when faced with unfamiliar cases while unprofessional doctors may ask for some irrelevant symptoms that cause patients' distrust. To quantify whether a DSMAD agent is reliable, we introduce two new evaluation criteria for reliability, namely, Internal Trust (Int.) and External Trust (Ext.).

**Internal trust.** In order to model the internal trust of a diagnosis system, we adopt the bootstrapping method to produce multiple diagnosis results from 100 models, $m^{(k)}, k \in [1, 100]$. We employ a random sampling strategy with replacement to generate training set for each model. The models are initialized with different random parameters. By considering the noises of data sampling and parameter initialization, the expected diagnosis prediction of the bootstrapping models ought to be less sensitive to these noises. We calculate the expectation of the diagnoses from these models by:

$$\mathbf{Pr}_{\text{Int.}}(d|p) = \mathbf{Pr}_{\{m^{(k)}\}}(d|p) = \mathbf{E}_{\{m^{(k)}\}}[m^{(k)}(p) = d], \text{ and } \text{Int.}(m) = \frac{1}{D}\sum_{i=1}^{D}\mathbf{Pr}_{\text{Int.}}(m(p_i)|p_i).$$

$$(3)$$

where $p$ denotes a patient and d denotes a disease, $m$ is the target model and $D$ is the size of test patients.

**External trust.** As for modeling the external trust, we calculate the coverage ratio of the mentioned symptoms during the diagnosing process. Specifically, we first calculate the symptom-disease co-occurrence $\mathbf{Pr}_{\text{Ext.}}(\text{Symptom}|d)$ from the whole dataset. Then we calculate the external trust of each dialogue using the average probability of the mentioned symptoms corresponding to the ground-truth disease. Denote the set of symptoms inquired by the target model $m$ when interacting with the patient $p_i$ as $\{Symp._j, j \in [1, H_{p_i}]\}$, where $H_{p_i}$ is the number of the inquiries. Formally, the final external trust of the target model $m$ is the averaged external trust of all test dialogues as

$$\text{Ext.}(m) = \frac{1}{D} \sum_{i=1}^{D} \frac{1}{H_{p_i}} \sum_{j=1}^{H_{p_i}} \mathbf{Pr}_{\text{Ext.}}(\text{Symp.}_j|d_{p_i}^*), \tag{4}$$

where $D$ is the size of the test patients and $d_{p_i}^*$ is the ground-truth disease of patient $i$. In order to reach higher external trust, the system is required to inquire disease-relevant symptoms which are frequently inquired about in real life.

## 5.2 ROBUSTNESS

Robustness refers to the ability of a system to stay functioning correctly in the presence of noisy inputs or stressful environmental conditions. Enlightened by the adversarial attack theory of current machine learning models (Kurakin et al., 2018), in the context of the medical dialogue system, we consider the robustness of a model measured by the invariance under the noisy inputs which are designed to have little impact on the final diagnosis. Owing to the particularity of the medical diagnosis, that is, subtle changes in symptoms might lead to different diagnoses (for example, reversing the presence of a symptom might associate the patient with a complete distinct disease), we design two relatively innocuous patterns of noises to symptoms in a user goal: 1) Given the symptom-disease relationship, we move $0 \sim k$ symptoms with the weakest correlation to the ground-truth disease from explicit symptoms to the implicit symptoms; 2) Add $1 \sim h$ symptoms to the user goals that are mostly correlated to the ground-truth disease. To quantify the robustness of the model $m$, we calculate the unaltered ratio of the correct diagnoses given the noisy test set $P'$, formally:

$$\text{Rub.}(m; P') = \mathbf{E}_{P'}[m(p') = m(p)|m(p) = d^*], \tag{5}$$

where $p$ is the original data of the noisy data $p'$, and $d^*$ is the ground-truth disease. The robustness is calculated on the patients whose original diagnoses are correct (i.e., $m(p) = d^*$). This is because the misdiagnoses made by a robust system could be attributed to the inadequate symptom information from the patients. Therefore, a robust system is reasonable to change its diagnoses with more or less symptom information from these patients.

## 6 EXPERIMENTS

We begin by introducing the two open-released DSMAD benchmarks used in this paper and the baseline approaches. Then, we compare different methods w.r.t. diagnosis accuracy as well as our proposed metrics. Moreover, we have conducted a human evaluation of different approaches to better demonstrate the superiority of our method from the human expert perspective.

**Benchmarks.** In this paper, all experiments are conducted using the two open-released medical diagnosis datasets, i.e., MuZhi (MZ) and DingXiang (DX). MZ is a synthesized dataset proposed by Wei et al. (2018), which includes 586 training and 142 test records with 66 symptoms and 4 diseases; DX in Xu et al. (2019) consists of 423 training and 104 test records with 41 symptoms and 5 diseases[1]. Records in MZ are synthesized so that its records are clean and structural. Different from MZ, records in DX Xu et al. (2019) is collected in real life, so that the raw self-report corpus are maintained in these records. In both datasets, the diagnosis record contains the ground-truth disease, the explicit symptoms (i.e. the symptom information in self-report), the implicit symptoms (i.e. the symptom information mentioned during the discourse).

**Baselines.** In this section, we focus on evaluating and analyzing the performance of current DSMAD baselines, i.e., Basic DQN (Wei et al., 2018), KR-DS (Xu et al., 2019), including our proposed INS-DS. Basic DQN makes the first attempt to apply the classical reinforcement learning framework, deep q-network (Mnih et al., 2015), to dialogue-based medical automatic diagnosis system. Based

---

[1]The diseases in MZ are infantile bronchitis (Bronch.), upper respiratory tract infection (U.R.I.), infantile diarrhea (I.D.) and infantile dyspepsia (Dysp.); and the diseases for DX are allergic rhinitis (A.R.), upper respiratory tract infection (U.R.I.), infantile diarrhea (I.D.), infantile hand-foot-and-mouth disease (H.F.M) and pneumonia (Pneu.). For brevity, In the experiment section, disease names in the tables are abbreviations.

Table 1: Diagnosis accuracy of different baselines on MZ dataset and DX datasets

| Method | MZ | | | | | DX | | | | | |
|---|---|---|---|---|---|---|---|---|---|---|---|
| | I.D. | Dysp. | U.R.I. | Bronch. | Overall | A.R. | U.R.I. | Pneu. | H.F.M. | I.D. | Overall |
| SVM-ex | 0.89 | 0.28 | 0.44 | 0.71 | 0.59 | 0.5 | 0.92 | 0 | 0.8 | **0.95** | 0.64 |
| SVM-ex&im | 0.91 | 0.34 | 0.52 | 0.93 | 0.71 | 0.7 | 0.96 | 0.35 | 0.75 | 0.9 | 0.74 |
| Basic DQN (2018) | - | - | - | - | 0.65 | 0.7 | 0.79 | **0.55** | 0.7 | 0.9 | 0.73 |
| KR-DS (2019) | **0.96** | 0.39 | 0.5 | **0.97** | 0.73 | **0.9** | 0.67 | 0.3 | 0.95 | 0.9 | 0.74 |
| Our INS-DS | 0.87 | **0.55** | **0.60** | 0.82 | **0.73** | 0.75 | **0.96** | 0.3 | **0.95** | 0.8 | **0.76** |

Table 2: Dialogue performance on DX dataset

| Method | Acc. | Match rate | #turns |
|---|---|---|---|
| Basic DQN (2018) | 0.731 | 0.110 | 3.92 |
| Sequicity (2018) | 0.285 | 0.246 | 3.40 |
| KR-DS (2019) | 0.740 | 0.267 | 3.36 |
| Our INS-DS | **0.760** | **0.290** | 5.84 |

Table 3: Robustness on MZ dataset.

| Method | NS.1 | NS.2 | NS.3 |
|---|---|---|---|
| Basic DQN (2018) | 0.669 | 0.785 | 0.699 |
| KR-DS (2019) | **0.883** | 0.864 | 0.806 |
| KR-DS-relation* (2019) | 0.760 | 0.836 | 0.785 |
| Our INS-DS | 0.840 | **0.883** | **0.835** |

on this method, KR-DQN incorporates the pre-defined symptom-disease conditional probability to transform the output of DQN to improve diagnostic performance. For fair comparison, we adopt the same NLU from (Xu et al., 2019) for all baselines. Following the same settings as (Wei et al., 2018) and (Xu et al., 2019), we also includes the supervised learning baselines SVM-ex (using explicit symptoms as input), SVM-ex&im (using explicit and implicit symptoms as input) as well as Sequicity (Lei et al., 2018) (sequence-to-sequence model). Since these supervised learning methods were not trained or evaluated with interactive processes, we only evaluate the accuracy of these methods to highlight the effectiveness of the RL baselines.

**Training Details.** In general, the maximum discourse length of these baselines is 22. Rewards are critical to reinforcement learning. We follow the reward settings in (Xu et al., 2019), where +44 reward is used to encourage successful diagnosis and -22 to punish misdiagnosis. For each turn, -1 is used to penalize the failure to hit one of the recorded symptoms in the user goal. The philosophy of the scale of the reward is to balance the maximum accumulated rewards (+44) and minimum accumulated rewards (-22+-1×22=-44). Empirically, the size of the replay buffer is 1e6, the discount factor $\gamma$ is set as 0.95, the polyak factor $\alpha$ is 5e-3, the learning rate is 1e-4 and the optimization method is Adam (Kingma & Ba, 2014). The maximum interaction runs for each experiment is 1e6. At the end of the training, the lastest model with the best diagnosis accuracy on training set is adopted for evaluation. As for training 100 bootstrapping DSMADs and synthesizing noisy test datasets, we adopt different random seeds to keep them various.

As for the training data for the disease classifier, we use data in replay buffer which is collected along training. Specifically, at each training step, the RL agent interacts with the patient simulator and generates the dialogue state comprised of the symptom value vector and the binary visited indicator vector, as mentioned in Sec.3.1. At the end of each discourse, each of these dialogue states are paired with the disease label and stored into the experience replay buffer. When updating the disease classifier, a batch of training data is sampled from the replay buffer, and then the classifier takes the symptom value vectors and disease labels as the training inputs and labels, respectively. The classifier is training along the training of the RL policy.

## 6.1 QUANTITATIVE RESULTS

In this part, we compare different baselines with our method using the metrics diagnosis accuracy as well as our proposed DSMAD metrics for reliability and robustness. Note that, the results of the baselines are the best number reported in their papers if provided. Otherwise, the results are calculated by running the open/reproduced codes.

**Accuracy of diagnosis.** We compared the performance of our method and the state-of-art methods by diagnostic accuracy as well as matching rate, as shown in Tab. 2 and Tab. 1, which strictly follows the same settings as (Xu et al., 2019). Method Sequicity (Lei et al., 2018) is only evaluated on DX because it requires raw language as input. And the results of each disease of Basic DQN in Tab. 2 is missing because the results are from the paper (Xu et al., 2019). According to these tables, our approach exceeds or reaches the comparable diagnosis accuracy of KR-DS which is guided by the external knowledge. In the Tab. 1, we observed that the accuracy of diseases "infantile diarrhea"

Table 4: Internal/external trusts and diagnosis entropy on MZ and DX datasets.

| Method | MZ | | | DX | | |
|---|---|---|---|---|---|---|
| | Int. | Ext. | Ent. | Int. | Ext. | Ent. |
| Basic DQN (Wei et al., 2018) | 0.6298 | 0.4054 | 0.6351 | 0.7428 | 0.0135 | 0.4563 |
| KR-DS (Xu et al., 2019) | 0.7999 | 0.4740 | 0.2204 | 0.7438 | 0.1632 | 0.3663 |
| Our INS-DS | **0.8508** | **0.4832** | **0.0414** | **0.7916** | **0.4327** | **0.0614** |

(I.D.) and "infantile bronchitis" (Bronch.) of baseline methods are much higher than the accuracy of the other two diseases. Aware of the similarity of these diseases ("infantile diarrhea" is similar to "infantile dyspepsia" and "infantile bronchitis" is similar to "upper respiratory tract infection"), we concludes that the baselines method have learnt a biased diagnoser to obtain a higher overall diagnosis accuracy. And our approach has achieved a more balanced performance. This implies our method is better at distinguishing two similar diseases instead of biasing to one of them. While on DX, Basic DQN has a more balanced results but its overall accuracy is lower. As shown in Tab. 2, our INS-DS has the most average interaction turns. This is because INS-DS is not allowed to jump into informing a disease until the introspective results are converged. The higher matching rate of INS-DS means that mentioned symptoms of INS-DS are more likely presented in the records.

**Reliability analysis.** We evaluated DSMAD baselines using reliability measures and the results are shown in Tab. 4. We found the internal trust of our model has obvious advantages over other methods (at least 0.048). This indicates that the diagnosis of our INS-DS is less sensitive to the noise during the training phrase, such as sampling error and parameter initialization. Basic DQN has the lowest internal trust on both benchmarks. To better understand the variance of diagnosis results, we also calculated the entropy (Ent.) of diagnoses from the bootstrapping models. Specifically, the entropy of diagnoses from the bootstrapping models for each test patient, and then the averaged entropy is reported in Tab. 4. Method with higher entropy indicates that the method is more likely to produce various diagnosis with training noise. Our INS-DS has the close to zero entropy, which also evidences that the diagnoses from INS-DS are more consistent and are more in line with the symptom-disease relation in the benchmarks. In terms of external trust, our model still achieves the highest score. Since the external trust is calculated based on the symptom-disease co-occurrence, it indicates that our INS-DS can inquire about more disease-relevant symptoms, so its diagnosis is more convincing to the patients. For Basic DQN and KR-DS, their external trust across MZ and DX are dramatically different. This is because the symptom-disease co-occurrence of DX is much denser than MZ's, so the values in the co-occurrence matrix of DX are generally smaller. And if the method does not inquires about those frequent symptoms on DX, the overall external trust calculated by averaging small probabilities is likely to be smaller.

**Robustness analysis.** A robust system is expected to maintain its correct diagnosis after inconsequential modification of its input. We set up three noisy test dataset according to the formulas proposed in Sec. 5.2. Specifically, (a) $NS.1$ consists of noisy samples constructed by randomly moving $0 \sim 2$ symptoms (which are mostly irrelevant with the ground-truth disease) from self-report to the implicit symptoms slots; (b) As for $NS.2$ and $NS.3$, we append 1 and 2 symptoms most related with the ground-truth disease to the explicit symptoms, respectively. We repeated the above process 10 times and calculated the averaged unaltered proportion of the correct diagnoses using Equ. (5). The results are presented in Tab. 3. We only evaluated the robustness on MZ dataset since the self-reports in DX are natural language [2]. As discussed in Sec.5.2, higher unaltered proportion means that the correct diagnosis is more robust to noisy information. We found that INS-DS achieved superior performance on these noisy datasets.

As for $NS.2$ and $NS.3$, the results of all approaches decreased when more disease-related symptoms are added. It indicates that current DSMAD methods are prone to memorize the pattern of the diagnosis dialogue record instead of learning the ground-truth symptom-disease relationship. This is also contradictory to our real-life experience, indicating the difference between current DSMAD methods and human cognition process. While by incorporating the introspective process, INS-DS could maintain a high invariance, which means that INS-DS are better at exploiting the ground-truth symptom-disease relation than other methods. It's worth noting that KR-DS also has good performance on these three settings since KR-DS is provided with a pre-defined symptom-disease knowledge graph. As we observed in Tab. 3, the performance of KR-DS without the graph,

---

[2]It's possible but time-consuming to revise the self-report of DX in order to evaluate the robustness on DX.

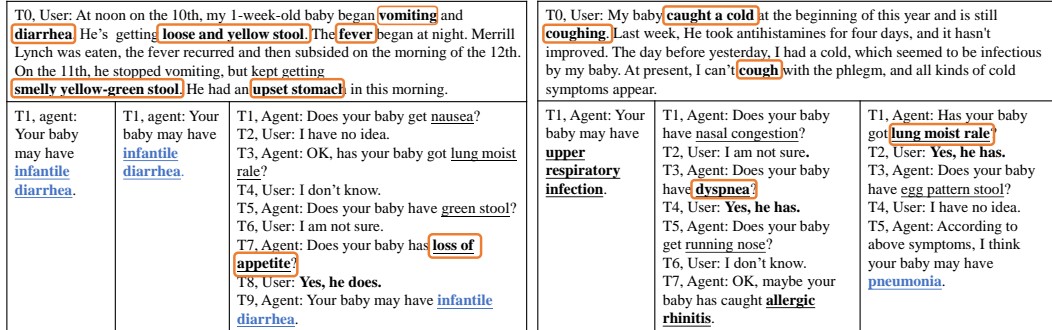

| T0, User: At noon on the 10th, my 1-week-old baby began **vomiting** and **diarrhea** He's getting **loose and yellow stool**. The **fever** began at night. Merrill Lynch was eaten, the fever recurred and then subsided on the morning of the 12th. On the 11th, he stopped vomiting, but kept getting **smelly yellow-green stool**. He had an **upset stomach** in this morning. | | | T0, User: My baby **caught a cold** at the beginning of this year and is still **coughing**. Last week, He took antihistamines for four days, and it hasn't improved. The day before yesterday, I had a cold, which seemed to be infectious by my baby. At present, I can't **cough** with the phlegm, and all kinds of cold symptoms appear. | | |
|---|---|---|---|---|---|
| T1, agent: Your baby may have **infantile diarrhea**. | T1, agent: Your baby may have **infantile diarrhea**. | T1, Agent: Does your baby get nausea? T2, User: I have no idea. T3, Agent: OK, has your baby got lung moist rale? T4, User: I don't know. T5, Agent: Does your baby have green stool? T6, User: I am not sure. T7, Agent: Does your baby has **loss of appetite**? T8, User: **Yes, he does.** T9, Agent: Your baby may have **infantile diarrhea**. | T1, Agent: Your baby may have **upper respiratory infection**. | T1, Agent: Does your baby have nasal congestion? T2, User: I am not sure. T3, Agent: Does your baby have **dyspnea**? T4, User: **Yes, he has.** T5, Agent: Does your baby get running nose? T6, User: I don't know. T7, Agent: OK, maybe your baby has caught **allergic rhinitis**. | T1, Agent: Has your baby got **lung moist rale**? T2, User: **Yes, he has.** T3, Agent: Does your baby have egg pattern stool? T4, User: I have no idea. T5, Agent: According to above symptoms, I think your baby may have **pneumonia**. |

Figure 2: Two visualized conversation results on DX dataset of basic DQN, KR-DS and our INS-DS methods. In each table, the first line is a self-report of the patient. The resulted dialogues from the left to right are Basic DQN, KR-DS and INS-DS respectively. The correct mentioned symptoms are highlighted with orange boxes, and the correct diagnosis is blue underlined.

named KR-DS-relation*, is severely degraded. Since the policy of INS-DS without the introspection module has the similar structure as Basic DQN, we conclude that the introspection module enables INS-DS to better perceive the symptom-disease dependencies and obtain the similar performance with KR-DS which has the pre-defined symptom-disease relationship.

## 6.2 QUALITATIVE RESULTS

For an intuitive recognition of baselines, we provide some visual results, as shown in Fig. 2. The first line of each running example is the self-report of the patient and the second line is dialogues of DQN, KR-DS, and INS-DS consecutively. Basic DQN often gives a reckless diagnosis without inquiring about more symptoms. KR-DS did make the attempts to inquire some symptoms but have limited ability to grasp useful symptoms. Our INS-DS performs better in acquiring relevant symptoms and is more cautious in conclusion with more reasonable interactions.

It is not sufficient to evaluate a dialogue system merely by automatic metric, we invited three students, who are in pursuit of a medical doctorate, to perform human evaluations based on three aspects: 1) *diagnosis validity*, the correctness of final diagnosis; 2) *symptom rationality*, the rationality of requested symptoms based on the medical knowledge; 3) *topic transition smoothness*, the smoothness of the inquiry logic irrelevant to the disease. Different from the diagnosis accuracy in Sec.6.1, participant were asked to score the diagnoses validity not only referring to the ground-truth disease but also the symptom-inquiry process. We generated dialogues of the test dataset by using each DSMAD baseline incorporated with NLU and NLG modules from Xu et al. (2019). Then we asked the participants to score each of the randomly

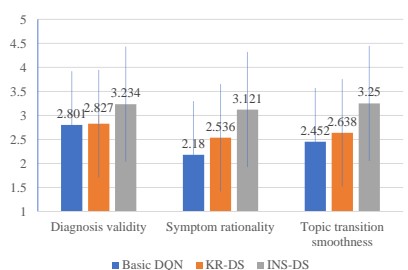

Figure 3: Results of human evaluation on DX dataset, our INS-DS has exceeded in diagnosis validity, symptom rationality and topic smoothness.

shuffled dialogues (to ensure a double-blind rating process) from 1 to 5 (integer). At the end, we received 104*3=412 assessments for each index and for each approach. The larger the score, the better the dialogue. The averaged scores and standard deviations are shown in Fig. 3. From the perspective of human experts, our INS-DS exceeds the other two methods in the three standards, especially in the aspect of symptom rationality and topic transition smoothness. The standard deviations are about 1.1[3]. The standard deviation is large since the scores are discrete integers.

## 7 CONCLUSION

In this work, we heuristically designed the INS-DS model to mimic the rational decision-making process in real life. Our approach achieves superior performance in different evaluation metrics, including our proposed reliability and robustness metrics. Experimental results evidence that our method can better understand the symptom-disease relation and therefore is insensitive to other irrelevant noisy factors.

---

[3]From the left to right of the scores in Fig. 3, standard deviations are 1.10, 1.11, 1.09, 1.06, 1.24, 1.16, 0.95, 0.87 and 1.04, respectively

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
