# OpenReview forum: "Towards a Reliable and Robust Dialogue System for Medical Automatic Diagnosis"
_ICLR.cc/2021/Conference — Reject_

### Official Review · AnonReviewer4 · 2020-10-27
**INS-DS shows improvement over existing DSMADs, but clarifications are needed regarding experimental settings**

**Rating:** 6
**Confidence:** 4

**Review:**

In this manuscript, authors proposed a novel dialogue system for medical automatic diagnosis (DSMAD) called INS-DS. There are three components in the general DSMADs, which are NLU, DM and NLG. In this paper, NLU and NLG components are adopted from Xu et al., 2019. Authors focused on designing decision-making parts in the dialogue management (DM) component. INS-DS includes two modules: an inquiry module and an introspective module which is inspired by the introspective process when humans make decisions. Authors also introduced two metrics evaluating the reliability and robustness of general DSMAD agents.

Authors demonstrated INS-DS achieved state-of-the-art performance compared with other DSMAD agents and performed better with respect to robustness and reliability. My concerns are mainly related to the experimental settings and the detailed comments are listed as follows.

#### Major Comments:
1. (Table 4) Evaluating and improving the robustness and reliability of DSMAD agents is one of the major contributions in this manuscript. Authors have evaluated the robustness and reliability of DSMAD agents based on proposed metrics. One of the further experiments authors can conduct is to check the performance of DSMAD agents trained on MZ or DX and test on the other. This is a general and widely-accepted approach to demonstrate the agents' robustness and reliability.

 Both DX (527 conversations) and MZ (710 conversations) are relatively small datasets and there are only two diseases (I.D. and U.R.I) that appeared in both of the datasets. The performances regarding I.D. are pretty consistent but this is not true for U.R.I. The symptoms related to U.R.I. may be different across these two sets but there should be a reasonable overlap. Could you train the model on DX and test it on MZ focusing on I.D. and U.R.I? This would be more convincing to demonstrate the agent's robustness and reliability and this helps understand the inconsistency of diagnosis accuracies regarding U.R.I.

2. (Table 1) For Basic DQN and KR-DS, their performances in Ext. (External trust) across MZ and DX are dramatically different. Is there any reason that can explain this difference especially considering that DX dataset was proposed along with KR-DS?

3. (Table 2) It's better to include the baseline performances (test dataset without noise) for each of the agents. Otherwise it is hard to see whether agents are robust to noise or not under different settings (NS.1, NS.2 and NS.3 ).
4. (Table 3). If more related symptoms are appended, the tasks should be relatively easier for humans. But NS.3 is consistently worse than NS.2 across all agents. Could you elaborate on this observation?

5. (Figure 3) How does the diagnosis validity correlate with the ground-truth diagnostics for the test set? The diagnosis validity score ranges from 2.8 to 3.2 which is far from the best score 5. Is this due to the inconsistency between the ground-truth diagnosis and the diagnosis made by students?

6. (Page 7, robustness analysis) In the robustness analysis, authors made use of noise test sets to demonstrate the agents' robustness. For humans, the diagnosis should be invariant with respect to the orders of the explicit symptoms or implicit symptoms. Based on this assumption, it is natural to augment the train set by permuting/sampling symptoms and this should further improve the performances of all models. Have you applied this augmentation strategy in the training phase?


#### Minor Comments:
1. (Figure 3) It is helpful to include the error bars (or all three data points) to show the variance of scores assigned by students.

2. (Table 2) Robusteness -> Robustness

---

> ### Author Response · Authors · 2020-11-17
> **Add more analysis of the experiments**
>
> Thank you for your constructive suggestion and instructive comments on the analysis of our experiments.
>
> __Q: The reviewer suggested evaluating the cross-benchmark performance to indicate robustness and reliability.__
> However, we found that none of the DSMAD methods in our manuscript is applicable. The reasons are two-fold: a) The two benchmarks might have different diagnosis logic since MZ is a synthesized dataset and DX is a real-world dataset; b) the types of symptoms are mismatched in the different dataset so that even if the experiments were conducted with two common diseases (I.D. and U.R.I), the action spaces of the agents with different dataset are different. Therefore, in our manuscript, instead of proposing cross-benchmark metrics, we propose in-benchmark metrics to take robustness and reliability into consideration. We do appreciate the suggestion of the reviewer and it’s a valuable direction for the future study.
>
> __Q: Explain the sharp contrast between MZ and DX on the external trust.__
> The external trust on DX is smaller because the values in the co-occurrence of DX is smaller. The average entropy of probabilities of symptom w.r.t. disease of MZ is 5.206 and the average entropy of DX is 8.600. This implies that the probabilities of symptoms are more averaged on DX. Therefore, on DX, the average of small probabilities w.r.t. inquired symptoms is prone to be smaller than that of MZ.
>
> __Q: Provide the robustness result of clean data__
> The robustness metrics is the unaltered ratio of correct diagnoses of the original test set, as elaborated in Sec.5.2. The robustness values in Tab.3 are calculated relatively to the performance on the original test set.
>
> __Q: As for NS.2 and NS.3, analyze the results of all approaches that decreases as more disease-related symptoms are added.__
> It might imply that current DSMAD methods are prone to memorize the pattern of the diagnosis dialogue record instead of learning the ground-truth symptom-disease relationship. This is also contradictory to our real-life experience, indicating the difference between current DSMAD methods and human cognition process. While by incorporating the introspective process, INS-DS could maintain a high invariance, which means that INS-DS are better at exploiting the ground-truth symptom-disease relation than other methods.
>
> __Q: Provide more details on human evaluation.__
> We have elaborated the details of human evaluation in our revised manuscripts. We invited three students, who are in pursuit of a medical doctorate, to perform human evaluations. Different from the diagnosis accuracy in Sec.6.1, participants were asked to score the diagnoses validity not only referring to the ground-truth disease but also the symptom-inquiry process. The participants were asked to score each of the randomly shuffled dialogues from 1 to 5 (integer). In the end, we received 104*3=412 assessments for each index and for each method. We found that the participants were more likely to give a score from 2~4 instead of giving the scores 1 and 5. the score, resulting in the averaged score distant from 1 and 5.
>
> __Q: Do the experiment augment the training data via permuting/sampling?__
> In our experiment, such augmentation is not explicitly applied to train the DSMAD methods. However, since the training data for DSMAD is collected by the RL agent, a similar permuting/sampling augmentation process actually happens during repeated trial and error of the RL agent. Specifically, at each step, an RL agent has a chance to inquire about a random symptom and updates the state according to the response. In this sense, the RL agent indeed acts as a sampler and generates permuted/sampled states along the training process.
>
> __Q: There are some minor typos.__
> We have corrected the typos/confusing sentences mentioned by the reviewer. Thank you for your kind reminders.

---

### Official Review · AnonReviewer3 · 2020-10-28
**The authors propose an interactive RL based medical dialog system with a novel introspection mechanism**

**Rating:** 4
**Confidence:** 4

**Review:**

##########################################################################

Summary:

The authors propose a new method for an RL agent-based , interactive medical dialogue. The method has two core components: a symptom inquiry module which selects a new symptom to inquire about a time step t, and an introspection module which looks ahead one time step to evaluate *if* knowing the outcome of this inquiry will change the predicted disease of the patient at time step t + 1. If the predicted disease at t + 1 would not change on the basis of issuing the symptom query at time t, the system returns the diagnosis, else it issues the query and repeats this process until convergence. The authors argue that this mimics the human decision making process and leads to a more principled diagnostic procedure.

The authors also propose two new metrics by which to evaluate medical dialogue systems. Reliability, which measures how sensitive the model is to random fluctuations during training (internal) and how well the selected symptoms align with human doctors (external). Robustness, which measures how well the model performs under an application-specific adversarial attack.

##########################################################################

Rationale for score:

I proposed a recommendation of reject due mainly to the deep concerns I have about the experimental setup. The two example vignettes the authors have provided are both seriously flawed both with respect to the accuracy of the ground truth (symptoms and diagnoses) and with respect to the example dialogues (see below). These issues will substantially impact at least one of novel metrics proposed by the authors (external reliability), which arguably is the most important metric. Due the experimental issues, I am unable to trust this metric and, unfortunately, the authors' method is no better (and in some cases substantially worse) on the second most important metric (in my assessment), diagnostic accuracy.

##########################################################################

Pros:

1. The proposed approach is novel and interesting and addresses an important problem. I particularly like the introspection aspect. I do however wonder if authors could provide some theoretical insight into situations where their one-step look ahead procedure will and will not converge? e.g. is the proposed method always guaranteed to terminate after a finite number of steps or are there situations when it might never return a diagnosis without a hard, pre-defined step cutoff?

2. The figure is clear and extremely helpful.

3. The introduction of new metrics other than diagnostic accuracy is new and somewhat well motivated.

##########################################################################

Cons:

1. There seems to be very little theoretical connection between internal and external reliability, as described. It might be better to break these out into two different metrics.

2. Internal reliability is not fully defined. From equation (3), it appears to be the average probability from k=100 bootstrapped models. That, to me, seems to be a way to *reduce* aleatoric uncertainty but does not measure model sensitivity to perturbations. Something like the standard deviation of this bootstrapped probability would measure that however.

3. Even though the authors are using a pre-existing dataset, I have deep concerns about how realistic the simulated datasets are, and how it may specifically impact their proposed metrics. For example, I showed the conversation about infantile diarrhea to my clinical colleague who is a practicing neonatologist and board certified pediatrician. Several things that were indicated to be the correct symptoms (orange boxes) were completely nonsensical in her assessment. In her opinion, no physician would ever describe a baby who has trouble feeding as "anorexic" and a "stomach murmur" is not a term she has ever heard. Likewise in the second example her opinion is that none of the systems collected sufficient information to render a diagnosis of pneumonia, and that actually upper respiratory infection (agent 1) provides the most sensible diagnosis, and the symptom requested by INS-DS (egg pattern stool) makes little sense.  Unfortunately, if these examples are representative of the rest of the data, it casts doubts on the strength of the authors claims. Their external reliability metric rests on how well the ground truth symptoms are captured by a system, and it seems that there could be systemic issues with the ground truth symptoms and diagnoses. The authors report some human assessment with three medical students, but it is hard to understand how the evaluation was done due to insufficient detail, especially in light of the two examples provided in the paper.

4. Unfortunately, the authors proposed approach is not as accurate as other methods in terms diagnostic accuracy, and sometimes by a substantial margin. As the authors argue, other factors are certainly important (reliability and robustness), but all are secondary to diagnostic accuracy in my opinion. If the system you are interacting with is robust to noise that's great, but if it performs worse on average, then you will do worse in the long run as a potential user.

Minor typos/confusing sentences:
- "Dialogue system for medical automatic diagnosis (DSMAD) aims at learning an agent to collect patient’s information and make preliminary diagnosis"

- "but ignore the importance of robustness and reliability for practical medical application"

- "Despite the contradictory diagnoses, the agent will still get a positive reward from the upper one since the diagnosis is correct." (what is the upper one?)

- "resulting in the system vulnerable to the noise happened during the interactions."

- "As for evaluation, aware of the insufficient validity of final classification accuracy and the number of query steps, current methods measure the performance of the diagnosing process by computing the hitting rate of the inquired symptoms."

- "Therefore, measures DSMAD according to the recorded symptoms in each user goal could not reflect
the actual performance of DSMAD in general."

- " Different from the most popular diagnosis systems which laving equal emphasis on symptom-inquiry and disease-diagnosis"

- ". And only when the most valuable inquiry proposed by the symptom-inquiry module makes no difference to the future diagnosis will the predicted disease be imformed"

- "Emperically, the size of the replay buffer is 1e6,"

- "Trusts for reliability. " (Section header, should be tests?)

---

> ### Author Response · Authors · 2020-11-17
> **Examples are rational; and INS-DS has a better overall accuracy of diagnosis**
>
> Thank you for your constructive comments and helpful suggestion on our experiment. The major concern from the reviewer is based on the "seriously flawed" examples. However, we respectfully disagree with the reviewer’s comments on our dialogues examples as well as the diagnosis accuracy of our method.
>
> __Q: The reviewer described the rationale of his concerns referring to the opinions of his clinical colleague.__
> 1) __For the first opinion, "several things that were indicated to be the correct symptoms (orange boxes) were completely nonsensical",__ we did some search on some health care websites, and we found proof of the reality of our two examples. For the first example, "Seattle Children’s" healthcare website proposes a tutorial about infantile diarrhea [Link][1]. In this tutorial, "vomit", "loose stools", "stools are yellow", "fever", "stools … smells bad" are mentioned as symptoms of infantile diarrhea. And "stomach is upset" and "loss of appetite" are mentioned in "Healthline"[Link][2]. For the second examples, "cold" and "cough" are two common symptoms for both "upper respiratory infection" and "pneumonia", as described in two tutorials from "Healthline"[Link][3] and [Link][4]. And the most distinction between upper respiratory infection and pneumonia is that they happen in different parts of the human body. As described in "Cystic Fibrosis News Today"[Link][5], pneumonia is a kind of lower respiratory (including lung) infections. And the "lung moist rale" is an indicator of lung disease as well as a typical symptom of pneumonia[Link][6]. Our INS-DS actually does a good job in finding the most valuable symptom to inquire about (i.e. lung moist rale) and makes the correct diagnosis (i.e. pneumonia). Compared to ours, Basic DQN (agent 1) made a rushy and wrong diagnosis (i.e., upper respiratory infection).
>
> [1]: https://www.seattlechildrens.org/conditions/a-z/diarrhea-0-12-months
> [2]: https://www.healthline.com/health/diarrhea-and-loss-of-appetite
> [3]: https://www.healthline.com/health/acute-upper-respiratory-infection
> [4]: https://www.healthline.com/health/pneumonia
> [5]: https://cysticfibrosisnewstoday.com/2017/07/18/upper-respiratory-tract-infections-vs-lower-respiratory-tract-infections/
> [6]: https://en.wikipedia.org/wiki/Respiratory_sounds
>
> 2) __For the opinion "no physician would ever describe a baby who has trouble feeding as "anorexic" and a "stomach murmur" is not a term she has ever heard",__ we realize this problem is attributed to language translation. Both MZ and DX are in Chinese. And the examples are translated from Chinese to English for demonstration. We have rewritten the examples in our revised manuscript. Thank you for your helpful comments.
>
> 3) __As for the "egg pattern pool" raised by INS-DS in the second example,__ the possible reason could be that our INS-DS might learn a strategy to exclude the other candidate diseases by inquiring about the most distinctive symptoms.
>
> __Q: Unfortunately, the authors' proposed approach is not as accurate as other methods in terms of diagnostic accuracy.__
> As for the diagnosis accuracy, our method is better than all of the baselines in terms of the overall diagnosis accuracy, since the DSMAD agent is trained to obtain the highest overall performance, as shown in Tab.1. On MZ dataset, our INS-DS obtain a comparable overall diagnosis performance as KR-DQN. However, our method obtains a more balanced accuracy for each disease. More analysis is presented in our revised manuscript.
>
> __Q: Provide some theoretical insight into situations where their one-step-look-ahead procedure will and will not converge.__
> One can view the proposed module as the tree-pruning function, which actually decreases the searching space of the policy. Theoretically, our proposed method is promised to stop. Because in the worst case, if all symptoms are visited, introspective results converge as there are no more so-called one-step-look-ahead states. In contrast, for DSMAD methods whose RL policy is responsible for deciding when to stop, there is a chance that the RL policy will never choose to stop. However, in practice, our proposed method is more likely to inquire about more symptoms to meet the introspection condition, as depicted in Tab.2 of our revised manuscript. We believe being conservative is always better than being reckless in the field of medical application.
>
> __Q: Something like the standard deviation of this bootstrapped probability would be more proper to measure sensitivity to perturbations.__
> Actually, incorporating the proposed internal trust metric, we also evaluate the entropy of diagnoses of the bootstrapping models. The results are presented in Tab.4 (i.e., Ent.).
>
> __Q: Minor typos/confusing sentences__
> We have corrected the typos/confusing sentences mentioned by the reviewer. Thank you for your kind reminders.

---

### Official Review · AnonReviewer2 · 2020-10-28
**Good paper, need more clarity on some aspects**

**Rating:** 6
**Confidence:** 4

**Review:**

This paper proposes an interesting medical diagnosis dialogue system. It considers cooperation between two separate modules: an inquiry module that manages symptom inquiries and an introspective module to decide when to inform the diagnosis based on available symptoms. The paper also proposes two evaluation metrics based on reliability and robustness of the models. Results show the usefulness of the proposed model in terms of these evaluation metrics along with a human evaluation study.

Strengths:

- The idea of considering the symptom inquiries and the diagnosis decision making as a separate cooperative process in a deep reinforcement learning setting is interesting.

- The proposal of how to meaningfully evaluate the proposed models using two new metrics is very useful.

- Experimental results appear to be solid and the comparisons with related work are also meaningful.

Weaknesses:

This paper appears to be an incremental piece of work based on the prior work on the same task. In my opinion, the main limitation of the paper is its novelty in terms of the core idea as it claims to mimic the decision making process of the human in defining the introspective system - however, this idea and the framework presented in Figure 2 is not new (see Ling et al. "Learning to Diagnose: Assimilating Clinical Narratives using Deep Reinforcement Learning"). Furthermore, the core modules of the paper are essentially same as Xu et al. 2019 as also acknowledged by the authors in Section 3.1.

Few other comments:

- More clarity needed for the notion of contradictory diagnosis and correct diagnosis as they appear to be ambiguous.

- Not sure about the rationale behind discriminating the symptoms between explicit and implicit - why was it necessary to consider them differently for modeling?

Overall, the proposed methodology is solid and the work would be beneficial to the community.

---

> ### Author Response · Authors · 2020-11-17
> **Clarify the difference of our method from the priors**
>
> Thank you for your constructive comments and help us to extend the related works. Actually, our work has fundamental differences to the prior works, from either the framework or evaluation metrics.
>
> The most fundamental difference is that our method introduces an introspection module to provide a more reliable "stop mechanism", that’s, the RL agent is only allowed to stop inquiring about symptoms if the future information makes no difference. Different from ours, prior works rely on the policy agent to decide when to stop in a black-box manner. Furthermore, we also proposed two novel metrics for evaluating the reliability and robustness of different methods.
>
> __Q: The reviewer recommended an interesting paper, "Learning to Diagnose: Assimilating Clinical Narratives using Deep Reinforcement Learning", which also draws inspiration from the human cognition process to improve the performance of diagnosing.__
> Briefly, Ling et al. propose a model to select evidence from the external clinical source according to the hint sentence of the target clinical document, and then the RL agent is adopted to filter out the useless evidence. However, this work is still significantly different from ours. From the perspective of tasks, Ling et al. aim to improve the diagnosis of clinical documents instead of dialogue, and the RL agent of Ling is not used to interact with users. From the perspective of motivation to our understanding, the work proposed by Ling et al. is more related to the memory recall [1] and selective attention mechanisms [2] in the human decision-making process, instead of the introspection like ours. From the perspective of the framework, Ling et al. include an external clinical source for searching candidate evidence and apply RL to determine whether to collect the evidence. Since they do not include the introspection process, the RL agent is also required to decide when to stop. In this perspective, the pipeline proposed by Ling et al. is more similar to the methods of Basic DQN and KR-DQN.
>
> __Q: The NLU and NLG modules are not novel.__
> As for a medical application, the decision-making logic is more critical than the NLU and NLG modules. Therefore, in our manuscript, we focus on improving the performance of the decision-making part in dialogue management. And all DSMAD methods in our experiment use the same NLU and NLG modules for a fair comparison.
>
> __Q: Adjust the content (e.g. “contradictory diagnosis”) in the introduction section for better clarity.__
> Thank you for your kind reminder. We have adjusted the content in the introduction section.
>
> __Q: What’s the difference between explicit and implicit symptoms?__
> The explicit symptoms are symptoms presented in the self-report of the diagnosis dialogue record. And the implicit symptoms are symptoms mentioned during the discourse between the patient and the doctor. Generally, an explicit symptom is proposed by the patient at the beginning of the discourse, while the implicit symptom is proposed by the doctor in order to collect more evidence. In this sense, the distributions of the implicit and the explicit symptom are different. From the perspective of RL, explicit symptoms form the initial states. Therefore, in our manuscript, we still follow the settings in KR-DQN, that’s, the explicit states are presented in the self-report.
>
>
> [1] Richards, B. A., & Frankland, P. W. (2017). The Persistence and Transience of Memory. Neuron, 94(6), 1071–1084.
>
> [2] Treisman, A. M. (1964). Selective attention in man. British medical bulletin, 20(1), 12-16

---

### Official Review · AnonReviewer1 · 2020-10-28
**Review of Towards a Reliable and Robust Dialogue System for Medical Automatic Diagnosis**

**Rating:** 6
**Confidence:** 4

**Review:**

The paper proposes a method for automatic medical diagnosis in a dialog system which is comprised of two modules: one which proposes symptoms to inquire about, and another which decides whether to go ahead with the inquiry or inform a disease. The second module makes the decision by looking ahead and checking whether the symptom wold case differences in determining the disease. The authors argue that existing systems have only been evaluated with respect to diagnosis accuracy, and introduce two additional metrics to evaluate the reliability and robustness of the method. They evaluate their method and compare with existing state-of-the-art methods on two datasets. Furthermore, they do a small human evaluation and report results on how evaluators perceive the diagnosis validity, symptom rationality, and topic transition smoothness of different methods. Overall, the paper is well motivated and written, and the authors show equal or better results compared to other state-of-the-art methods.

I have the following questions/comments. Given clarifications in an author response, I would be willing to increase the score.
- Can you add more information about the datasets/augmented test sets used (e.g., train/dev/test size). What data is the disease classifier trained on? What do the diseases in Table 4 represent?
- Some details about the training are missing. For example, how many seeds were used? How long does it take for the training to converge using each method?
- For the results of the DX dataset, is the same NLU model used for all three models? If not, how does that impact the results?
- What do the numbers in Table 2 indicate? Are they accuracy results on the noisy test-sets?
- The section "Accuracy of diagnosis" is not well written. Sequicity is introduced without any discussion on what it is. Additionally, new methods are introduced at the end of the paragraph. Why are these methods not used in Table 1? What is the intuition behind different results for different diseases? I suggest rewriting this section to first introduce the methods that are used, and then discussing the results. It might make sense to put this section before the "Trusts for reliability" section on page 6.
- In the same section, the sentence "And the results of each disease of Basic DQN is missing because the results are from the paper", it is not clear that refers to table 4.
- Since you discuss Figure 2 in detail in section "Qualitative Analysis", I suggest removing it from section "Accuracy of diagnosis" as mentioning it just briefly in this section is confusing.
- There is no discussion of the errors that the proposed approach is making. Are the errors the same as the ones made with the other methods?

The text has spelling errors:
 - Page 3, P2: measures DSMAD -> measuring
 - Page 3, P3: dataset is includes -> dataset includes
 - Page 3, P4: paper is the same -> paper are the same
 - Page 4, P1: record -> records, produce -> produces, Remove And from the beginning of the sentence
 - Page 4, P2: imformed -> informed
 - Page 8, P1: an reckless -> a reckless (what does a reckless diagnosis refer to?)
 - Page 8, P2: ask -> asked. Remove And from the beginning of the sentence
 - Page 8, P3: mimics -> mimic, could -> can, therefore insensitive -> therefore is insensitive

---

> ### Author Response · Authors · 2020-11-17
> **Reorganizing the experiment section to improve the clarity.**
>
> Thank you for your comprehensive comments and constructive suggestions. We have thoroughly reorganized the experiment section to improve clarity. In the experiment section, we first introduce the benchmarks and baselines used in our experiments. Then, we describe the details of the training. After that, we present the quantitative comparison results w.r.t. diagnosis accuracy, reliability and robustness. Finally, we present the qualitative results of different methods.
>
> __Q: Provide more details about the benchmarks.__
> The two open-released DSMAD benchmarks used in our paper are MuZhi (MZ), a synthesized dataset, and DingXiang (DX), a real-world dataset. MZ is composed of 586 training and 142 test records with 66 symptoms and 4 diseases; DingXiang (DX) composed of 423 training and 104 test records with 41 symptoms and 5 diseases. The diseases in MZ are infantile bronchitis, upper respiratory tract infection, infantile diarrhea and infantile dyspepsia; and the diseases for DX are allergic rhinitis, upper respiratory tract infection, infantile diarrhea, infantile hand-foot-and-mouth disease and pneumonia. In both datasets, the record contains the ground-truth disease, the explicit symptoms presented in the self-report, and the implicit symptoms mentioned during the dialogue. These two datasets are in Chinese and we translated them into English for demonstration. To measure the reliability, new training data for each bootstrapping model are sampled with replacement using different random seeds (i.e. 100 random seeds). As for measuring the robustness, we synthesize three new noisy test datasets each with ten times the size of the original test dataset.
>
> __Q: Add more training details.__
> We have further elaborated on the training details, including the training data for the disease classifier. The training data for the disease classifier is collected during the training process. Specifically, during training, at each time-step, the RL agent interacts with the patient simulator and generates a dialogue state which includes a symptom value vector. At the end of each diagnosis, the dialogue states generated during the interaction are paired with the disease label and stored into the experience replay buffer. When updating the disease classifier, the classifier takes the symptom value vectors and the diseases as the training input and label from the replay buffer, respectively. The number of training iterations for each experiment is set as 1e6, which costs about half a day on a PC with CPU i7-9700K. After training, the latest model with best training diagnosis accuracy is adopted for evaluation.
>
> __Q: For the results of the DX dataset, is the same NLU model used for all three models?__
> The same NLU and NLG modules are used for all baselines, i.e., Basic DQN, KR-DQN and our INS-DS.
>
> __Q: What do the numbers in Table 2 indicate?__
> The number in Tab.3 (Tab.2 originally) indicates the unaltered proportion w.r.t. the correct diagnoses of the original test set. The unaltered proportion is formally defined in Sec.5.2. The rationale of using the unaltered ratio to indicate the robustness of a DSMAD system is that: a robust system should not change its decision given the negligible noise. The higher unaltered proportion indicates the system is more robust to the noise.
>
> __Q: Suggest rewriting "Accuracy of diagnosis" section and adding more analysis on the accuracy.__
> As kindly suggested by the reviewer, we have rewritten the section  “Accuracy of diagnosis" and put it before “Reliability Analysis". As for the diagnosis accuracy of each disease, INS-DS achieves a more balanced and superior performance compared to the other baselines. For more analysis, please refer to our revised manuscript. As for the supervised learning baselines, e.g. Sequicity, a sequence-to-sequence method, we have described them in the new paragraph "baselines". Unlike Basic DQN and KR-DQN, these supervised learning baselines do not include the interaction process during training. Therefore, following the same settings as KR-DQN, we only compare them in diagnosis accuracy to demonstrate the benefit of reinforcement learning.
>
> __Q:  Are the errors the same as the ones made with the other methods?__
> We found that our method is more likely to fail when the symptom information is not enough to satisfy the introspection module. For example, our INS-DS inquires about "egg pattern stool", "loose stool’, "stomach bloating" from a patient with infantile diarrhea. However, due to the missing information problem inherent in the passive observational patient record, the patient simulator replies to these inquiries by "not-sure". These cases usually lead our method to fail. Therefore, as the interaction persists longer (i.e. more difficult the introspection condition is met), the more likely our method is going to fail.
>
> __Q: The text has spelling errors.__
> Thank you for your kind reminder. We have corrected the spelling errors in our revised manuscript.

---

### Author Response · Authors · 2020-11-25
**To all reviewers**

Thank you for your constructive comments. We have adjusted our manuscript accordingly.

According to the comments from all reviewers, we concluded the three major concerns/interests from the reviewers.

1) __Adding more experiment details__

2) __Elaborating on the analysis of the experimental results__

3) __The validity of our demonstration examples__

As for the third concern raised by reviewer3, there is substantial medical evidence ([Link][1], [Link][2], [Link][3], [Link][4], [Link][5], [Link][6]) to justify the validity of our examples. Some "unusual" expressions in the examples are attributed to language translation, and we have corrected them. We believe the major concern causing the reviewer3 vote for rejection has been well-addressed in our comment.

In conclusion, addressing the reviewers' suggestions and concerns, we have made significant clarifications to the paper, while striving to keep the broad ideas and findings intact. In summary, the following changes have been made:

1) We have thoroughly reorganized the experiment section to improve clarity. In the experiment section, we first introduce the benchmarks and baselines used in our experiments. Then, we describe the details of the training. After that, we present the quantitative comparison results w.r.t. diagnosis accuracy, reliability, robustness. And finally, we present the qualitative results of different methods.

2) According to the comments of reviewers, we have extended the analysis of our experimental results.

3) We have rewritten the demonstration examples with often-used expressions.



[1]: https://www.seattlechildrens.org/conditions/a-z/diarrhea-0-12-months
[2]: https://www.healthline.com/health/diarrhea-and-loss-of-appetite
[3]: https://www.healthline.com/health/acute-upper-respiratory-infection
[4]: https://www.healthline.com/health/pneumonia
[5]: https://cysticfibrosisnewstoday.com/2017/07/18/upper-respiratory-tract-infections-vs-lower-respiratory-tract-infections/
[6]: https://en.wikipedia.org/wiki/Respiratory_sounds

---

### Decision · Program_Chairs · 2021-01-07
**Final Decision**

**Decision:**

Reject

**Comment:**

The authors propose a novel approach to a dialog-based  automated medical diagnosis, and present promising empirical results. The focus of this work is on robustness and reliability besides just the accuracy of diagnosis, which appears to be an important aspect in medical applications. The paper is clearly written and well-motivated. However, in there are still several concerns raised by the reviewers, and the paper may require a bit of extra work to be ready for publication.